# Melatonin in Combination with Albendazole or Albendazole Sulfoxide Produces a Synergistic Cytotoxicity against Malignant Glioma Cells through Autophagy and Apoptosis

**DOI:** 10.3390/brainsci13060869

**Published:** 2023-05-27

**Authors:** Miguel Hernández-Cerón, Víctor Chavarria, Camilo Ríos, Benjamin Pineda, Francisca Palomares-Alonso, Irma Susana Rojas-Tomé, Helgi Jung-Cook

**Affiliations:** 1Doctorate in Biological and Health Sciences, Universidad Autónoma Metropolitana, Mexico City 04960, Mexico; miguelqfbuamx@gmail.com (M.H.-C.); crios@correo.xoc.uam.mx (C.R.); 2Neuroimmunology and Neuro-Oncology Unit, Instituto Nacional de Neurología y Neurocirugía (INNN), Mexico City 14269, Mexico; vchavarria@innn.edu.mx (V.C.); benjamin.pineda@innn.edu.mx (B.P.); 3Laboratorio de Neurofarmacología Molecular, Departamento de Sistemas Biológicos, Universidad Autónoma Metropolitana, Unidad Xochimilco, Mexico City 04960, Mexico; 4Neuropsycopharmacology Lab, Instituto Nacional de Neurología y Neurocirugía, Mexico City 14269, Mexico; francisca.palomares@innn.edu.mx (F.P.-A.); isrtome@hotmail.com (I.S.R.-T.); 5Pharmacy Department, Universidad Nacional Autónoma de México, Mexico City 04510, Mexico

**Keywords:** glioblastoma, melatonin, albendazole, albendazole sulfoxide, synergism

## Abstract

Glioblastoma is the most aggressive and lethal brain tumor in adults, presenting diffuse brain infiltration, necrosis, and drug resistance. Although new drugs have been approved for recurrent patients, the median survival rate is two years; therefore, new alternatives to treat these patients are required. Previous studies have reported the anticancer activity of albendazole, its active metabolite albendazole sulfoxide, and melatonin; therefore, the present study was performed to evaluate if the combination of melatonin with albendazole or with albendazole sulfoxide induces an additive or synergistic cytotoxic effect on C6 and RG2 rat glioma cells, as well as on U87 human glioblastoma cells. Drug interaction was determined by the Chou–Talalay method. We evaluated the mechanism of cell death by flow cytometry, immunofluorescence, and crystal violet staining. The cytotoxicity of the combinations was mainly synergistic. The combined treatments induced significantly more apoptotic and autophagic cell death on the glioma cell lines. Additionally, albendazole and albendazole sulfoxide inhibited proliferation independently of melatonin. Our data justify continuing with the evaluation of this proposal since the combinations could be a potential strategy to aid in the treatment of glioblastoma.

## 1. Introduction

Glioblastoma (GB) is the most frequent malignant tumor of the central nervous system (CNS) in adults, and it has a poor prognosis. Currently, the standard treatment involves maximal surgical resection, followed by radiotherapy and chemotherapy; however, the median overall survival is between 12 and 15 months [1]. In 2017, bevacizumab, an angiogenesis inhibitor, received Food and Drug Administration approval for the treatment of adults with recurrent GB that has progressed following prior therapy; however, the median overall survival did not exceed 24 months [2]. The poor prognosis of GB treatment is related to the low specificity of chemotherapeutic agents, the difficulty of most antitumor agents to access the CNS due to the blood–brain barrier (BBB), as well as the limitation to intracellular accumulation of drugs in tumor cells mediated by efflux transporters [3,4,5]. Therefore, these challenges point to the need to develop new therapies for this disease.

Drug repositioning has been a successful strategy to investigate existing drugs for additional clinical indications, with evidence supporting the anticancer effects of benzimidazole carbamates [6,7]. In this category, albendazole (ALB) has been addressed in different cancer models, including GB [8,9,10], with a well-tolerated high dose as anticancer treatment in clinical trials [11]. 

After oral administration, ALB is rapidly transformed into the chiral active metabolite albendazole sulfoxide [(+)-ALBSO; (−)-ALBSO)], which possesses anthelmintic activity, and into the non-chiral metabolite albendazole sulfone, which lacks pharmacologic activity. Studies from microsomal investigations in several species suggests that CYP3A4 and flavin-containing monooxygenase (FMO) are major enzymes responsible for the formation of sulfoxide metabolites from ALB [12]. Lee et al. reported that the ALBSO formation from ALB is also mediated by the CYP2J2 isoform, and significantly higher than those by the CYP3A4 isoform [13]. ALBSO readily crosses the BBB due to its high lipid solubility, presenting high availability in CNS, with almost half the concentration in cerebrospinal fluid than in plasma [14,15]. ALB and ALBSO are classically known for their affinity for tubulin and alteration of the microtubule assembly [16]. In addition, ALB has been reported is pleiotropic drugs with multiple effects on cells, including the inhibition of phosphorylation signaling pathways [17] and induction of oxidative stress promoting DNA fragmentation [18].

Melatonin (MLT), an endogenous indolamine synthesized primarily by the pineal gland, regulates numerous processes in humans, such as the sleep–wake cycle, immunomodulation, and endocrine function. MLT is primarily metabolized to 6-hydroxymelatonin, but MLT can also be deacetylated to 5-methoxytryptamine and N^1^-acetyl-5-methoxykynuramine in the CNS, with antioxidant properties to capture reactive oxygen species and reactive nitrogen species [19,20,21]. MLT is a highly lipophilic molecule that can diffuse through the cell membrane to interact with intracellular targets [22]. Several studies have shown the potential use of MLT in the treatment of cancer [23], including GB, with synergistic activity when combined with other drugs, attributed to the inhibition of multiple pro-survival pathways, the inhibition of efflux pumps, and the regulation of autophagy [24,25,26,27].

The present study was performed to evaluate if the combination of MLT with ALB or with ALBSO induces an additive or synergistic cytotoxic effect in glioma cells. Likewise, the cell death mechanisms involved were investigated. The assays were conducted on three of the most widely used cell lines (C6, RG2, and U87). 

## 2. Materials and Methods

### 2.1. Reagents, Drugs, and Antibodies

Dulbecco’s Modified Eagle Medium (DMEM), antibiotic-antimycotic solution (10,000 units of penicillin, 10 mg of streptomycin, and 25 μg of amphotericin B per mL), 10× trypsin solution, albendazole (ALB), albendazole sulfoxide (ALBSO), melatonin (MLT), propidium iodide (PI), crystal violet, and 3-(4,5-dimethylthiazol-2-yl)-2,5-diphenyltetrazolium bromide (MTT) were obtained from Sigma-Aldrich (St. Louis, MO, USA). Fetal bovine serum (FBS) was obtained from Biowest (Nuaillé, Pays de la Loire, France). MACS bovine serum albumin was obtained from Miltenyi Biotec (Bergisch Gladbach, Germany). APC-Annexin V Apoptosis Detection Kit with 7-AAD was obtained from BioLegend (San Diego, CA, USA). Dimethyl sulfoxide (DMSO) and ethanol (Merck, Readington Township, NJ, Germany) were of analytical reagent grade. Acridine orange (AO) was obtained from Polysciences (Warrington, PA, USA). The goat polyclonal antibody anti-MAP LC3 was obtained from Santa Cruz Biotechnology (Dallas, TX, USA), and the anti-goat IgG-FITC antibody was obtained from Abcam (Cambridge, UK).

### 2.2. Glioma Cells and Cell Culture

C6 and RG2 rat malignant glioma cell lines and U87 human glioblastoma cell line were acquired from the American Type Culture Collection (ATCC, Manassas, VA, USA). Cells were maintained in DMEM with 10% FBS and 1% antibiotic-antimycotic solution in a 37 °C incubator with 5% CO_2_ atmosphere and 98% relative humidity. Cells were maintained in culture flasks until they reached 80–90% confluence. Confluent cells were washed with phosphate-buffered saline (PBS) and detached by incubation in 1× trypsin solution, for collection and seeding.

### 2.3. Concentration-Effect and Combination Study

For concentration-effect study, the stock solutions of ALB 2000 μM and ALBSO 20,000 μM were prepared in DMSO. Additionally, a stock solution of MLT 200 mM was prepared in ethanol. The stock solutions were serially diluted in DMEM to prepare working solutions of each drug to obtain final concentrations 0.16, 0.24, 0.36, 0.55, 0.83, and 1.25 μM for ALB; 2, 4, 8, 16, 32, and 64 μM for ALBSO; and 0.18, 0.37, 0.75, 1.5, 3, and 6 mM for MLT. DMSO and ethanol concentrations in DMEM did not exceed 0.5% and 3%, respectively. Solutions of DMSO and ethanol were used as vehicle control. To evaluate the cytotoxic effect of the treatments, 3 × 10^3^ cells were seeded into 96-well tissue culture plates. Then, 24 h later, the cells were incubated with 100 μL of working solutions of ALB, ALBSO, MLT, and vehicle. After 72 h of treatment, the medium was removed, and cells were washed with PBS, and then 100 μL of MTT solution at a concentration of 5 mg/mL in DMEM was added to each well and incubated for 3 h at 37 °C. Afterward, the medium was aspirated, and blue formazan crystals were solubilized with 100 μL of DMSO. Absorbance was determined using a microplate reader (Synergy LX, BioTek, Winooski, VT, USA) at 570 nm. Six replicates were evaluated for each treatment, and the experiments were repeated at least four times. The cell viability percentage was calculated by the formula:(Absorbance of treated group/Absorbance of vehicle) × 100

The median dose effect (Dm) equivalent to mean inhibitory concentration (IC_50_) of the concentration–response curves was calculated using the Chou–Talalay method [28,29] and CompuSyn.exe^®^ software (Version 1.0), developed from the physical–chemical principle of the mass-action law analysis via mathematical induction and deduction.

Once the Dm values of each drug were calculated, they were used to design the combination study. The experimental procedures for preparing the solutions and assessing the cell viability were the same as described in the concentration-effect study. DMEM with maximum 0.5% of DMSO and 3% of ethanol was prepared as vehicle control. Each experiment was performed in triplicate over six repetitions. The combination index (CI) was calculated from the Chou–Talalay method using CompuSyn.exe^®^ software (Version 1.0), which represents a quantitative measure of the extent of drug interaction with the following ranges: CI = 0.1–0.90 (synergism), 0.90–1.10 (nearly additive), and 1.10 to >10 (antagonism) [28,29].

### 2.4. Determination of Cell Death Mechanisms

The combinations selected for the study were those that presented the greatest cytotoxic effect. In the C6 cell line, the concentrations were ALB 0.6 μM-MLT 0.6 mM and ALBSO 20 μM-MLT 1 mM, while for the RG2 cell line, the concentrations were ALB 0.6 μM-MLT 0.6 mM and ALBSO 26 μM-MLT 0.9 mM. For the U87 cell line, the concentrations used were ALB 0.45 μM-MLT 0.45 mM and ALBSO 18 μM-MLT 0.45 mM. In addition, the effect of individual drugs at the same concentrations was evaluated. For the experiments, 2 × 10^4^ cells were seeded into 24-well tissue culture plates. Then, 24 h later, the cells were incubated with 1 mL of working solutions of the combinations and vehicle. After 48 h of treatment, the culture medium was transferred to flow cytometry tubes, and the cells were washed with PBS. Then, cells were detached by adding 1X trypsin solution and were harvested into the same centrifuge tubes. The samples were centrifuged at 2000 rpm for 5 min, and the supernatants were discarded, taking care not to throw the button of sedimented cells.

#### 2.4.1. Apoptosis Detection with Annexin V and 7-AAD Double Stain

To detect annexin V bound to phosphatidylserine (PS) in the extracellular plasma membrane and 7-AAD bound to DNA, the Apoptosis detection with Annexin V and 7-AAD double stain assay was used [30]. To carry out these determinations, the treatments were prepared and processed as indicated in Section 2.4. After centrifuging and removing the supernatant, the pellet was resuspended with APC-labeled Annexin V and 7AAD in 100 μL of binding buffer. After 15 min of incubation at room temperature in the dark, 400 μL of binding buffer was added to analyze the cells by flow cytometry within 1 h after treatment. A total of 10,000 events were acquired in a FACS Calibur flow cytometer (BD Biosciences, Franklin Lakes, NJ, USA). Analysis was performed using CellQuest Pro and FlowJo v10 software. The dot plots were divided in quadrants to quantify the viable cells (Q4: Annexin V−/7AAD−), total apoptotic cells (Q3: early apoptosis, Annexin V+/7AAD− plus Q2: late apoptosis, Annexin V+/7AAD+) and necrotic cells (Q1: Annexin V−/7AAD+). The fluorescence distribution was shown as a colored dot plot analysis. Data were obtained from three independent experiments performed in triplicate.

#### 2.4.2. Evaluation of Autophagy

##### Detection of Acidic Vesicular Organelles

Autophagy is characterized by the formation and promotion of acidic vesicular organelles (AVOs) [31]. We used the lysosomotropic agent acridine orange (AO), which moves freely across biological membranes when it is uncharged; its protonated form accumulates in acidic cell compartments, where it forms aggregates that fluoresce bright red, as we have previously reported [32]. Flow cytometry with AO staining was employed to detect and quantify the cells with AVOs. In AO-stained cells, the cytoplasm and nucleus fluoresce bright green and dim red, respectively, whereas acidic compartments fluoresce bright red. Therefore, we measured the change in the intensity of the red fluorescence to obtain the percentage of cells with AVOs. To carry out these determinations, the treatments were prepared and processed as indicated in Section 2.4. Briefly, after centrifuging and removing the supernatant, cells were resuspended and stained with 300 μL of a solution of 1 μg/mL AO in DMEM for 15 min at room temperature and analyzed on a CytoFlex SRT cell sorter (Beckman Coulter, Brea, CA, USA), measuring the green (FL-1, *x*-axis) vs. the red (FL-3, *y*-axis) fluorescence of AO in a linear scale. Dot plots are divided in quadrants, where the sum of the upper-left and the upper-right quadrants of the dot plot (red fluorescent events) was used to represent the percentage of autophagic cells. These assays were performed in triplicate.

##### LC3 immunofluorescence Staining

The microtubule-associated protein 1 light-chain 3 (LC3) is essential for amino-acid starvation-induced autophagy and is associated with the autophagosome membrane [33]. In this case, 1.5 × 10^4^ glioma cells were seeded on chamber slide dishes (BD Biosciences, Franklin Lakes, NJ, USA), and treated with the drug concentrations and vehicle indicated in Section 2.4. After 48 h of treatment, cells were fixed with cold methanol for 30 min, washed twice with PBS and blocked with 2% bovine serum albumin for 10 min three times. After that, cells were incubated with the goat polyclonal antibody anti-MAP LC3 (1:400) for 30 min at room temperature. Then, cells were washed twice with PBS, blocked with 2% bovine serum albumin three times for 10 min, and incubated by additional 30 min in darkness with an anti-goat IgG-FITC antibody (1:400), washed again with PBS, and finally mounted with DAPI-mounting fluid. Images were obtained on a Leica DMLS microscope, with a 100× objective using the Leica Application Suite software (v. 4.0).

### 2.5. Proliferation Assay

For this test, the crystal violet dye was used, which binds to proteins and DNA molecules of attached cells, where cell proliferation can be calculated in relation to the amount of biomass present after treatment, since dead cells are shed, reducing the staining with crystal violet [34]. Briefly, 2 × 10^3^ cells were seeded into 96-well plates and treated with the drug concentrations indicated in Section 2.4. Media were removed, and cells were fixed with a solution of cold ethanol (70%) for 30 min at room temperature after 1, 3, 5, and 7 days of treatment. Finally, cells were stained with a crystal violet solution (0.1%) for 30 min, and the supernatants were discarded. Crystals were dissolved with 100 μL of 10% glacial acetic acid solution, and absorbance was measured in a spectrophotometer Eon at 570 nm. The relative cell proliferation of each treatment group was calculated by the formula:Absorbance of “X” group on each day (day 1, 3, 5, 7)/Absorbance of “X” group on day 1
where the “X” group represents a specific treatment group, dividing the absorbances obtained on days 1, 3, 5, and 7 by the absorbances of the same treatment group on day 1, individually, obtaining the relative cell proliferation. Analysis of the proliferation assay was performed by comparing the relative cell proliferation between the treatment groups on each day.

### 2.6. Statistical Analysis

Data were expressed as the mean ± standard deviation (SD). GraphPad Prism 6 software (v. 6.07) was used for statistical analysis, normality of the data was assessed with the Kolmogorov–Smirnov test, and statistical analysis was performed with the Kruskal–Wallis test followed by a Dunn’s multiple comparison test.

## 3. Results

### 3.1. ALB, ALBSO, and MLT Induced a Cytotoxic Effect on C6, RG2, and U87 Cell Lines

We found that all three drugs induced a cytotoxic effect in a concentration-dependent manner (Figure 1). In addition, the Dm values for ALB were 0.6 μM, 0.6 μM, and 0.9 μM in the C6, RG2, and U87 cell lines, respectively. For the ALBSO, the Dm values were 20 μM, 26 μM and 36 μM in the C6, RG2, and U87 cells, respectively. In the case of MLT, the obtained Dm values were 1 mM, 0.9 mM, and 0.9 mM, for C6, RG2, and U87 cells, respectively.

### 3.2. The Combination of MLT with ALB or ALBSO Induced a Synergistic Cytotoxicity

Due to the ability of ALB, ALBSO, as well as MLT to induce cytotoxicity in the C6, RG2, and U87 cells, we tested whether the combination of ALB with MLT and ALBSO with MLT could induce an additive or synergistic cytotoxic effect. Based on the Dm of each drug, we combined ALB with MLT in a 1:1 ratio concentration for all cell lines. In the case of the combination of ALBSO with MLT, the ratio concentrations were 20:1 for C6 cells, 29:1 for RG2 cells and 40:1 for U87 cells. The results showed that most of the combinations caused a higher percentage of cytotoxicity than the single drugs. According to the Chou–Talalay method, most of the combinations of ALB-MLT caused a synergistic cytotoxic effect (CI < 1.00). In the case of ALBSO-MLT, most of the combinations caused a synergistic cytotoxic effect in the C6 and U87 lines, while in the RG2 line, this synergy was found only in the combinations with the highest concentrations (Figure 2).

### 3.3. Effect of the Combinations of MLT with ALB or ALBSO on the Cell Death Mechanisms

#### 3.3.1. The Combinations of MLT with ALB or ALBSO Induced Apoptosis

Regarding the evaluation of the mechanisms involved in the decrease of tumor cell viability, our results showed that the combination of ALB 0.6 μM-MLT 0.6 mM induced apoptosis in 36% of C6 cells, statistically higher compared to 4.3% produced by the vehicle (*p* < 0.01), while MLT and ALB induced apoptosis only in 9.4 and 22.8% of cells, respectively (Figure 3a,d). Similarly, the combination of ALBSO 20 μM-MLT 1 mM induced a statistical increase in apoptotic C6 cells, with a mean of 43.4% (*p* < 0.01).

In the RG2 cell line, the combination of ALB 0.6 μM-MLT 0.6 mM induced apoptosis in 26.7% of cells, statistically higher than the vehicle with 5.4% (*p* < 0.01), while MLT and ALB induced apoptosis in 7.8 and 18.7% of cells, respectively. While the treatment with ALBSO 26 μM-MLT 0.9 mM and ALBSO 26 μM alone induced similar percentages of apoptotic cells, with 16.2 and 16.1% (*p* < 0.01), respectively (Figure 3b,e).

In the U87 cells, the combination of ALB 0.45 μM-MLT 0.45 mM increased the apoptotic cells, with 17.1% (*p* < 0.01), while the vehicle produced 7.1%, MLT alone produced 12.4%, and ALB alone produced 15.1% of apoptotic cells. Likewise, the combination ALBSO 18 μM-MLT 0.45 mM showed similar percentages of apoptosis, with 15% of apoptotic cells (*p* < 0.01) (Figure 3e,f). Regarding the percentage of necrotic cells, there were no statistical differences between groups.

#### 3.3.2. The Combinations of MLT with ALB or ALBSO Induced Autophagy

ALB has been reported to induce autophagy in human colon adenocarcinoma cells [35]; thus, we evaluated the contribution of autophagy to the cytotoxicity induced by the drug combinations. First, we verified the formation of LC3 puncta by immuno-fluorescence microscopy of glioblastoma cells. As seen in Figure 4a, the C6 cells show a higher expression of LC3 and the formation of the LC3 punctuate pattern after the treatment with ALB 0.6 μM, and in combination with MLT 0.6 mM, as well as when treated with ALBSO 20 μM, and in combination with MLT 1 mM. Similarly, the RG2 cells had a higher expression of LC3 and showed LC3 aggregation after the treatment with ALB 0.6 μM and in combination with MLT 0.6 mM, as well as when treated with ALBSO 26 μM and in combination with MLT 0.9 mM, as seen in Figure 4b. In addition, results in the U87 cells showed LC3 puncta formation after the treatment with ALB 0.45 μM and in combination with MLT 0.45 mM, as well as with ALBSO 18 μM treatment and in combination with MLT 0.45 mM, as seen in Figure 4c.

Next, we quantified the generation of AVOs in the tumor cells with AO staining by flow cytometry, indicating the percentage of cells with AVOs. In the C6 cells, the combination of ALB 0.6 μM-MLT 0.6 mM induced AVOs in 28.3% cells, statistically higher than the vehicle with 6.9%, MLT alone (7.8%), and ALB alone (17.1%) (*p* < 0.01), as seen in Figure 4d. Similarly, the combination ALBSO 20 μM-MLT 1 mM increased the C6 cells with AVOs, with a mean of 24.8%, statistically higher than MLT alone (12.4%) and ALBSO alone (17.9%) (*p* < 0.01). In the RG2 cells, the combination of ALB 0.6 μM-MLT 0.6 mM induced the highest percent of cells with AVOs (40.4%), statistically higher than the vehicle (9%), the MLT alone (14.1%), and the ALB alone (26%) (*p* < 0.01). Similarly, the combination of ALBSO 26 μM-MLT 0.9 mM induced a higher percent of RG2 cells with AVOs (37%), when compared to MLT alone (14%) and ALBSO alone (20.6%) (*p* < 0.01), as seen in Figure 4e. In the case of U87 cells, the combinations ALB 0.45 μM-MLT 0.45 mM and ALBSO 18 μM-MLT 0.45 mM induced a significative increase in cells with AVOs, showing values of 30.5% (*p* < 0.01) and 28.3% (*p* < 0.05), respectively, when compared to vehicle (13.7%), as seen in Figure 4f.

### 3.4. The Treatment with ALB and ALBSO Inhibited Proliferation, Independently of MLT

Then, we evaluated the proliferation rate of tumor cells by crystal violet staining during 7 days of treatment on C6 cells (Figure 5a), RG2 cells (Figure 5b), and U87 cells (Figure 5c). In addition, for the C6 cells, we found a significant suppression of proliferation from day 3 until day 7 of treatment with ALB 0.6 μM or ALBSO 20 μM, independently of the combination with MLT (*p* < 0.01), as seen in Figure 5d. A similar result was obtained in the RG2 cells, where the treatment with ALB 0.6 μM or ALBSO 26 μM suppressed the cell proliferation, finding a significant difference after day 3 when comparing both combinations of ALB-MLT and ALBSO-MLT to the vehicle (*p* < 0.05), as seen in Figure 5e. Figure 5f shows the proliferation rate of U87 cells, finding a significant suppression of cell proliferation after 3 days of treatment (*p* < 0.05) with ALB 0.45 μM or ALBSO 18 μM, despite the presence of MLT, showing a higher difference on days 5 and 7 (*p* < 0.001).

## 4. Discussion

Glioblastoma remains to be the most aggressive brain tumor in adults. Although in recent decades advances in the treatment of GB have been achieved, recurrence is often inevitable, and the survival of patients remains low; therefore, new treatment strategies are under evaluation, such as monoclonal antibodies, viral therapies, vaccines, drug repositioning, and drug combinations [36,37]. In recent years, the combination of drugs with different mechanisms of action is gaining more relevance in the treatment of GB, with the aim to increase the efficacy, lower drug doses, and counteract mechanisms of drug-resistance, among others [38].

In the present study, we evaluated the combination of ALB with MLT, since both drugs have demonstrated antitumor activity through different mechanisms of action [25,39]; therefore, we investigated if the combination could potentially synergize their antitumor effect. Likewise, we evaluated the combination of MLT with ALBSO, considering that ALBSO, the main metabolite of ALB, has shown the highest levels in plasma and cerebrospinal fluid after the oral administration of ALB [40]. The assays were conducted on three of the most widely used cell lines in the GB research, namely, the U87 human glioblastoma cell line and the C6 and RG2 rat malignant glioma cell lines, that have proven to be highly homologous to the GB [41].

Our results corroborate the cytotoxic effect that the ALB and MLT have on the C6 and U87 glioma cell lines [42,43,44,45]. In addition, the cytotoxic effect of these drugs on the RG2 line is reported for the first time. In all cell lines, MLT was the most effective and ALB the most potent. To date, there are few studies that have determined the IC_50_ values for ALB or MLT in glioma cells. Marslin et al. evaluated the cytotoxic effect of ALB in the U87 cell line and reported a value of 50.1 μM for ALB [10], which is higher than those found in the present study (0.9 μM). The difference could be related to the exposure time, since in our study the incubation time was 72 h, and in the previous study, it was performed at 24 h.

This is the first study demonstrating the antitumor activity of ALBSO against glioma cells. The only prior report of the antitumor activity of ALBSO shows the induction of apoptosis of breast cancer cells in vitro [46]. The antitumor activity of ALBSO is relevant given that it is highly available in the brain, so it could reach therapeutic concentrations in brain tumors such as GB. The results showed that this metabolite was as effective as ALB in inducing cytotoxicity against the glioma cells.

When the combination study was performed, we found that both ALB-MLT and ALBSO-MLT combinations produced an additive or synergistic cytotoxic effect on most combination ratios in all three glioma cell lines. It is worth noting that apoptosis was the main cell death mechanism associated with the treatments using the drugs alone and in combination, finding necrosis in a minimal percentage on the three glioma cell lines. Ehteda et al. found that the combination of ALB with 2-methoxyestradiol synergizes the induction of apoptosis of colon cancer cells and improves the survival of HCT-116 tumor-bearing nude mice [47]. The synergy was based on the sum effect of microtubule-binding activity of both drugs, which differs from our approach, which is based on the possible sum of the different mechanisms of action attributed to ALB and MLT, as mentioned above [17,18,24,27]. On the other hand, MLT also has shown a chemosensitizing effect, since MLT downregulates the expression of ABC transporter ABCG2, inducing the synergistic cytotoxicity when combined with TMZ against GB cells and GB-stem cells [26].

The formation of a punctuate pattern of LC3 is associated with the initiation of autophagy, via the aggregation of LC3 and the formation of the autophagosome, and directly correlated to the increase in AVOs in glioma cells, as an indication of the fusion of the autophagosome and lysosome [48]. Previous reports indicate that benzimidazoles, as mebendazole and ALB, can induce autophagy on GB cells [49] and colon adenocarcinoma cells [35], respectively, while the blockade of autophagy in cholangiocarcinoma cells, after its induction with ALB, has been associated with increased apoptosis of tumor cells [50]. Meanwhile, MLT has shown the ability to suppress autophagy in ovarian granulosa cells [51], as well as in rat brain neurons, through the reduction of reactive oxygen species [52]; however, the oxidative capacity of MLT metabolites has also been reported [53]. The autophagy in GB cells has been associated with the induction of cell death as a response to sustained cell damage, related to the accumulation of AVOs and the loss of the protective effect of autophagy [54]. In this regard, the interplay between the induction of apoptosis and autophagy is proposed to potentiate cell death in cancer cells and promote the effectiveness of anti-tumor molecules [55].

Previous reports show that the anti-proliferative effect of MLT is attributed to the suppression of miR-155 on U87 cells; however, these effects were found with very low concentrations of MLT (1 μM), compared to the concentrations used in this study [44]. MLT has shown a synergistic anti-proliferative effect when combined with sorafenib, by dual suppression of the STAT3 pathway in pancreatic cancer cells in vitro and in vivo [56]. In the case of ALB, its antiproliferative activity on C6 cells has been previously attributed to the inhibition of enzymes involved in the glycolytic pathway and lower ATP concentration in vitro and in vivo, showing an enhanced effect when ALB is loaded on silver nanoparticles [45]. In a similar way, thiabendazole, another antiparasitic benzimidazole, has proven to be effective at inhibiting proliferation of several GB cell lines by the downregulation of mini-chromosome maintenance protein 2 (MCM2) [57]. Shu et al. demonstrated that ALB plus Palbociclib, a cyclin kinase 4/6 (CDK4/6) inhibitor, synergistically suppresses melanoma cell proliferation in vitro and in vivo, by the dual arrest of cell cycle progression [58].

The gold-standard drug in the treatment of GB is temozolomide (TMZ); however, in vitro evaluations indicate the need for high concentrations, ranging from 100 μM to more than 1000 μM, to induce the desired effect on glioma cells [59]. There is evidence of the potentiation of the cytotoxic effect of TMZ in combination with other treatments against glioma cells, where apoptosis and autophagy can be synergized [60,61]; therefore, the evaluation of the combined effect of MLT-ALB/ALBSO to potentiate the antitumor effect of TMZ is proposed as a follow-up to this work.

New directions are needed for the combinations of ALB, ALBSO, and MLT. Recently, ALB has also been reported to promote immunotherapy response by facilitating ubiquitin-mediated PD-L1 degradation in melanoma models [62]; likewise, MLT has shown antitumor potential by impairing many of the characteristics that sustain cancer progression [63], highlighting the importance of discovering other potential mechanisms of action that could benefit the current treatment of patients with cancer.

Future perspectives include the evaluation in an in vivo model of orthotopic malignant glioma, which will allow us to evaluate the impact of the combined administration of these drugs on molecular markers, tumor eradication, and survival time, given that survival is a parameter of great importance to determine the therapeutic efficacy of a drug in the management of GB [64]. Potential in vivo studies could be based on the use of classical immunocompetent orthotopic malignant glioma models, as we have previously reported with C6 cells implanted in the brain parenchyma of Wistar rats [65], or as the model performed with GL621 mouse glioma cells in C57BL/6 mice [66].

## 5. Conclusions

The combined mechanisms of the pleiotropic drugs, ALB, ALBSO, and MLT, are relevant for the additivity and synergism found against the glioma cells. Considering the safety and inexpensive profiles of these drugs, and their high availability to the CNS, their combination could be a potential therapeutic strategy against GB. Other studies would be necessary to evaluate the antitumoral activity of these combinations in in vivo models.

## Figures and Tables

**Figure 1 brainsci-13-00869-f001:**
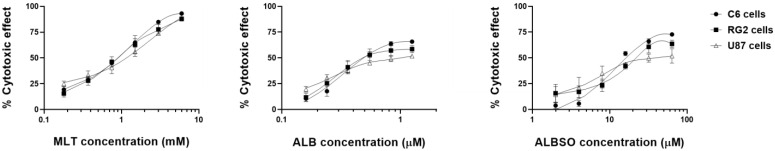
Concentration-effect curves of individual drugs. Cytotoxic effect after 72 h, in the three cell lines evaluated by the MTT reduction assay. Data obtained from four independent experiments, each with six replicates. Each dot represents mean ± SD.

**Figure 2 brainsci-13-00869-f002:**
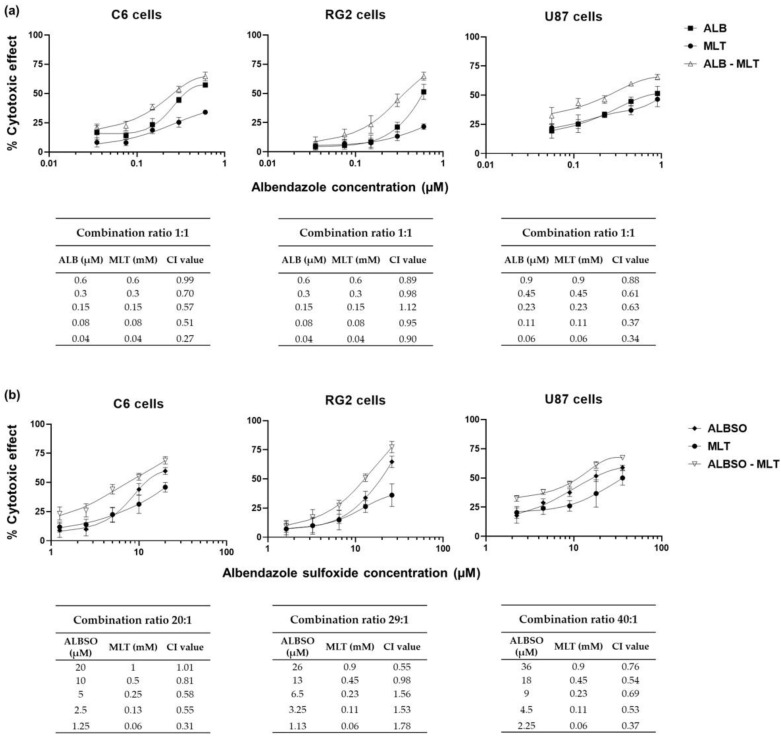
The combinations of MLT with ALB or ALBSO showed synergistic and/or additive effects. Graphic representation of the cytotoxic effect of the individual drugs and their combination, after 72 h of treatment, and the CI results in the three cell lines, for the combination of ALB with MLT (**a**) and ALBSO with MLT (**b**). Data obtained from three independent experiments in triplicate. Each dot represents mean ± SD.

**Figure 3 brainsci-13-00869-f003:**
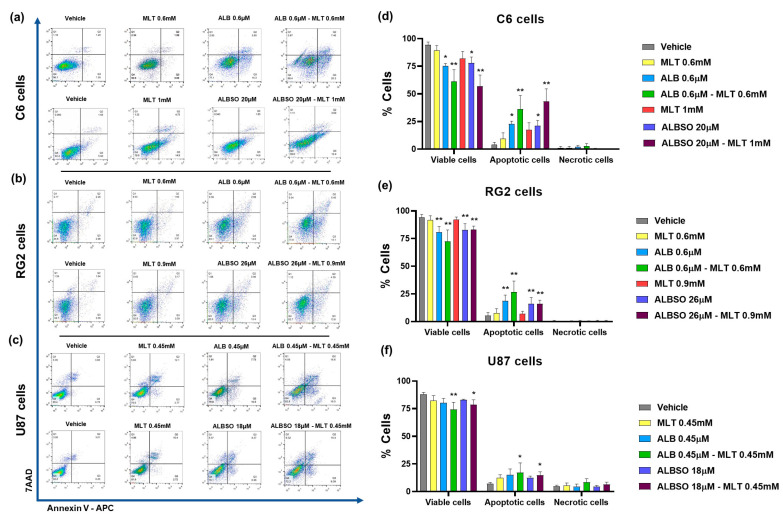
The combinations of MLT with ALB or ALBSO induced apoptosis. Representative dot plots and percentage of cells stained with Annexin V-APC/7AAD, quantified by flow cytometry after 48 h of treatment on C6 cells (**a**,**d**), RG2 cells (**b**,**e**), and U87 cells (**c**,**f**). The dot plots were divided in quadrants to quantify the viable cells (Q4: Annexin V-/7AAD-), total apoptotic cells (Q3: early apoptosis, Annexin V+/7AAD- plus Q2: late apoptosis, Annexin V+/7AAD+) and necrotic cells (Q1: Annexin V-/7AAD+). Data obtained from three independent experiments in triplicate (* *p* < 0.05, ** *p* < 0.01, compared to vehicle). Each bar represents mean ± SD.

**Figure 4 brainsci-13-00869-f004:**
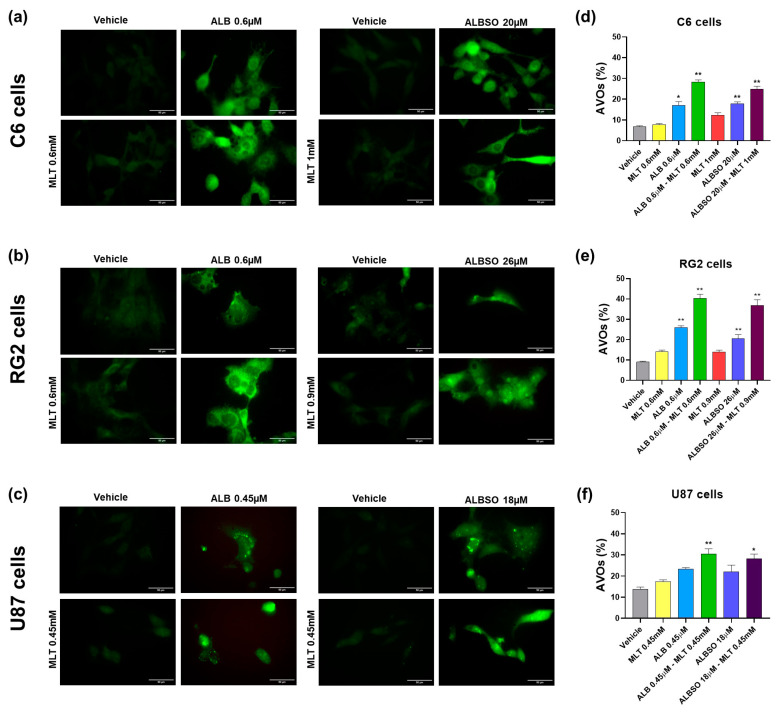
The combinations of MLT with ALB or ALBSO induced autophagy. Representative images of LC3-staining pattern by immunofluorescence and percentage of cells with AVOs quantified by flow cytometry after 48 h of treatment on C6 cells (**a**,**d**), RG2 cells (**b**,**e**), and U87 cells (**c**,**f**). Data obtained from three independent experiments in triplicate (* *p* < 0.05, ** *p* < 0.01, compared to vehicle). Scale bar equals 50 μm in microphotographs. Each bar represents mean ± SD in graphs.

**Figure 5 brainsci-13-00869-f005:**
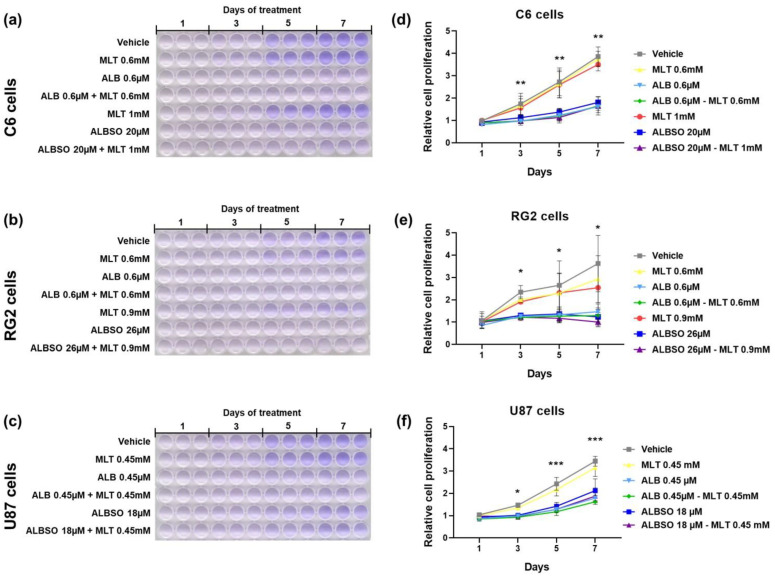
The treatment with ALB and ALBSO inhibited proliferation, independently of MLT. Representative images of crystal violet-stained cells in 96-well plates and graph with relative cell proliferation obtained after 1, 3, 5, and 7 days of treatment on C6 cells (**a**,**d**), RG2 cells (**b**,**e**), and U87 cells (**c**,**f**). Data obtained from three independent experiments in triplicate (* *p* < 0.05, ** *p* < 0.01, *** *p* < 0.001, compared to vehicle). Each dot represents mean ± SD.

## Data Availability

The data presented in this study are available on request to the corresponding author.

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
