# Peer review of "Melatonin in Combination with Albendazole or Albendazole Sulfoxide Produces a Synergistic Cytotoxicity against Malignant Glioma Cells through Autophagy and Apoptosis"

_brainsci, 2023, doi:10.3390/brainsci13060869_

Round 1

Reviewer 1 Report

1. There is a mistake in the introduction section (line 65).

2. Please provide whether each composition (mixture) induces additive or synergic effect using isobolographic analysis

3. The Authors should compare the results with a control compounds being clinically available drug used for the treatment of GB

4. Why did the Authors not performed the proliferation assay longer than 7 days? It would be nice to see whether this effect continues to increase or remains constant

moderate correction is required

Reviewer 2 Report

The manuscript entitled "Melatonin in Combination with Albendazole or Albendazole Sulfoxide Produces a Synergistic Cytotoxicity against Malignant Glioma Cells through Autophagy and Apoptosis" by Miguel Hernández-Cerón presents an investigation into the potential synergistic cytotoxic effects of melatonin in combination with albendazole or albendazole sulfoxide on glioma cells. Overall, the study provides valuable insights into novel therapeutic strategies for glioblastoma treatment.

Minor revision:

However, there are a few minor points that need to be addressed before the article can be considered for publication.

1.       It would be beneficial to include more specific details regarding the experimental methods employed, such as the concentrations and treatment durations for melatonin, albendazole, and albendazole sulfoxide. Additionally, it would be helpful to have comprehensive details of the methodology for flow cytometry, immunofluorescence, and crystal violet staining.

2.       The Chou-Talalay method was used to determine drug interactions and cytotoxicity. It would be useful to provide more details on this method, including a brief explanation of the principle and equations used. Additionally, the authors should specify the range of drug concentrations tested and the specific results obtained from the combination treatments. This information will enhance the transparency and reproducibility of the study.

3.       There are instances where sentences lack proper citations. For example, lines 185-186 state that the cells were treated with the same concentrations as previously mentioned. The authors should provide appropriate references to support these statements, ensuring that all information is properly attributed and traceable to the original sources.

4.       In the section titled "Detection of Acidic Vesicular Organelles," the authors are encouraged to provide a more detailed description of the methodology. This should include step-by-step instructions and reagent details. Elaborating on the methodology will facilitate the reproducibility of the experiments by other researchers.

5.       The conclusion suggests the need for further studies to evaluate the antitumoral activity of these combinations in in vivo models. It would be valuable to include a brief discussion on the potential design and considerations for future in vivo experiments. This discussion could encompass appropriate animal models, dosing regimens, and outcome measures to assess the therapeutic efficacy of the drug combinations.

Overall, the article provides important insights into the synergistic cytotoxic effects of melatonin in combination with albendazole or albendazole sulfoxide on glioma cells. By addressing the aforementioned points, the authors can significantly improve the clarity, reproducibility, and overall quality of the article, thereby strengthening its contribution to the field of GBM treatment.

Reviewer 3 Report

This manuscript presents the therapeutic efficiency of Albendazole + Melatonin and Albendazole Sulfoxide + Melatonin using 3 glioma cell lines. The subject is of interest and the manuscript is well structured.

Comments:

In the Introduction, please add more information regarding the metabolic pathway for Albendazole.

For Figure 4 a), b) and c), please enhance the quality of the images and add the magnification scale.

For Figure 4 d), e) and f), please use the same scale.

For Figure 5 d), e) and f), please use the same scale.
